# Evidence of a Threshold Size for Norwegian Campsites and Its Dynamic Growth Process Implications—Does Gibrat's Law Hold?

**Robin Valenta** [1], **Johannes Idsø** [2] and **Leiv Opstad** [1,*]

1   NTNU Business School, Norwegian University of Science and Technology, 7491 Trondheim, Norway; robinval@stud.ntnu.no

2   Department of Environmental Sciences, Western Norway University of Applied Sciences, 6856 Sogndal, Norway; Johannes.Idsø@hvl.no

*   Correspondence: leiv.opstad@ntnu.no

**Abstract:** Although campsites are an important segment of the tourist sector, few applied articles have analyzed their growth path and tested Gibrat's Law for firms within this industry. This knowledge can be of importance to the authorities when analyzing the regional impacts of growth in this sector. With government statistics from the last decade, we use a GMM framework to test the stricter version of Gibrat's Law, which consist of three parts: the campsites' growth trend, how they carry over success and failure, and how volatile their size is. The first and third part are rejected for Norwegian campsites, leading to a rejection of Gibrat's Law. To see if firms of different sizes follow different dynamics, we split the sample in three parts. Here, we find evidence of a threshold size, as large campsites follow a fundamentally different dynamic than small and medium campsites. Specifically, large campsites gain no stability in revenue by further increases in size, whereas they carry over success/failure across years. The opposite is true for the rest of the sector. Gibrat's Law is rejected on at least one count for each of the sub-samples. Lastly, we supplement the analysis with economy-wide and firm-specific variables to test further hypotheses.

**Keywords:** Gibrat's Law; campsites; tourism; growth; system GMM estimator; dynamic panel data; Norway



## 1. Introduction

In the growth path literature that uses Gibrat's Law, there are few studies that analyze campsites, and none that investigate Norwegian campsites. Previous studies of Italian and Dutch campsites do not reject Gibrat's Law, using an OLS framework on a sample of five years. This study uses a GMM (and ML) framework on a sample with twice the time dimension, although we have a smaller cross-sectional sample. The composition of the sample is also different, as Italian (Piergiovanni et al. 2003) and Dutch (Audretsch et al. 2004) campsites do not face the same degree of natural (and state) restrictions as Norwegian ones do, in addition to being larger than Norwegian ones. Unlike the previous literature, the present paper uses a detailed and accurate methodology to test Gibrat's Law. We hope to supplement the existing literature with our findings about the growth of businesses that work under these conditions, using a modernized framework.

The tourism industry is important for many countries and contributes to value creation and employment; accordingly, there is a relatively large amount of existing literature on the subject. Although campsites are an important contributor to this industry in Europe, North America, Australia, and parts of Asia, few analyses have tested Gibrat's Law for this segment.

It is of great interest to see if Gibrat's Law applies to the tourism industry in Norway, as Norway is moving towards a future in which it will be less dependent on oil. Therefore,

the focus must be on other industries The tourism industry is important for ensuring future sustainable regional development (NOU 2020). Therefore, the authorities need more knowledge about the campsites, among other things. For example, will the growth be distributed equally, or will there be a concentration around the large companies? The purpose of this article is to find out more about this, and related, issue(s).

The article in which Gibrat proposes the Law of Proportional Effect (LPE) (Gibrat 1931) has formed the basis for many research articles, and it states that an individual firm's relative growth is independent of firm size. Consequently, the best prediction one can make about any individual firm's size the next year will be that firm's current size, plus the growth in the relevant sector. If the year-to-year growth of large firms is 5%, it will be 5% for small firms as well. This does not mean that all firms grow at the same pace, but that the growth is independent of firm size.

The market concentration of industries and sectors is an essential topic in economics, thus how the distribution of market share changes over time is important. This dynamic is why the LPE has received so much attention, as it serves as the baseline with which to compare the growth dynamic in different industries and sectors. Any deviation from Gibrat's Law is evidence of the market at hand converging, at the extreme, towards perfect competition or monopoly. In most cases, when Gibrat's Law is rejected, it is rejected in favour of the mean reversion, although the very long run distribution of firms is often observed to be logarithmic rather than normal. This is due to the fact that sectors act more in accordance with Gibrat's Law the older they become.

## 2. Literature Review

Gibrat's Law has been an inspiration for many international publications (Daunfeldt and Halvarsson 2015) and many different methods and approaches have been applied. Mansfield (1962) applied Gibrat's Law in different ways. First, he tested if smaller firms were more likely to leave the market than larger firms. Then, based on economic theory, he investigated whether the companies had to pass a certain scale level, at which production exceeds the minimum efficient scale (MES) level, before Gibrat's Law holds. That is, it is possible that there is a threshold size at which a firm's growth pattern changes. Several other researchers have reported similar results, with the general conclusion being that larger firms grow independently of their size, as for the largest U.S. companies (Hymer and Pashigian 1962; Simon and Bonini 1958). The literature yields a mixed picture for industrial firms, as others show that this does not hold for small- and medium-sized enterprises (SMEs) (Becchetti and Trovato 2002; Hart and Oulton 1999; Fotopoulos and Louri 2004), whereas others show that the LPE holds for entire sectors (Buckley et al. 1984; Hymer and Pashigian 1962; Lensink et al. 2005; Simon and Bonini 1958). Most studies that reject Gibrat's Law show that the sector has a mean reverting tendency, meaning that smaller firms grows faster than larger firms (Almus 2000; Bartoloni et al. 2020; Daunfeldt and Elert 2013; Yadav et al. 2020). According to Jurado et al. (2021), Gibrat's Law applies to large capital-intensive companies that use advanced technology, which is taken to the extreme in other studies who suggest that large firms grow faster than their smaller competitors (Mukhopadhyay and AmirKhalkhali 2010). Finally, there are articles that did not find any hold for Gibrat's Law (Lotti et al. 2001).

Previous research has shown that Gibrat's Law applies to campsites based on data from the Netherlands (Audretsch et al. 2004) and Italy (Piergiovanni et al. 2003), meaning that it cannot be rejected that growth is independent of size. Furthermore, several authors have argued that Gibrat's Law applies to the service sector to a far greater extent than to manufacturing (Audretsch et al. 2004). The articles that failed in rejecting Gibrat's Law for campsites had rather short time dimensions of five years; however, this was in contrast to our ten years studied: from 2010 to 2019. This longer time dimension allows us to use the GMM framework to estimate the parameters of interest, whereas previous analyses of camping sites have used the less advanced OLS framework suggested by Chesher (1979). The choice of estimator is crucial when working with dynamic models using short panel

data, which is why we apply three different estimators that each have their strengths and weaknesses. Through this, we hope that we have attained results that are reliable, so that we can link them to economic theory with certainty.

If there is a minimum size a firm must obtain for survival (the MES), negative growth for small firms may result in deficits, which could, in the long term, lead to the closure of these firms (Audretsch et al. 2004). Mansfield (1962) reported that small firms have relatively higher death rates, but those that survive seem to have higher variation and grow faster than the big firms. He also noticed that firms with successful innovators grow twice as fast as others.

There is a substantial difference between the manufacturing and service industry (Audretsch et al. 2004). In the manufacturing industry, which depends heavily on capital as input, being small is obviously a drawback due to the economies of scale. If production exceeds the MES, the possibilities for profitable operation are far better. This might not be the case for the service industry, since the production is far less capital intensive. Consequently, there are less sunk costs, and economies of scale do not play as much of a key role as for manufacturing. This may explain why there are many small businesses in the service sector. There is a high proportion of family owned units in the Netherlands, and they often do not have ambitions to expand further. The same trend is also found in Italy, where a high proportion of companies in the hospitality sector have fewer than five employees (Piergiovanni et al. 2003). In Norway, the campsites are even smaller.

In a study of Italian hospitality industry (cafeterias, restaurants, cafes, campsites, and hotels), only cafeterias and campsites did not reject Gibrat's Law (Piergiovanni et al. 2003). An analysis based on Dutch data gave the same result for campsites (Audretsch et al. 2004). They state that growth in this sector is independent of firm size. For the four other sub-sectors within the field of hospitality, Gibrat's Law failed to hold. Park and Kim (2010) rejected Gibrat's Law for the restaurant industry, whereas Host et al. (2018) reported that the average growth of firms in the Croatian tourism sector was independent of their size. However, Ivandić (2015) did not confirm Gibrat's Law in a study of the hotel sector in Croatia, instead showing that the hotels tend to revert to a certain mean: smaller firms grew faster than the bigger ones. The growth was also shown to depend on ownership, where publicly owned companies had lower growth than privately owned ones.

## 3. Applying Gibrat's Law

The literature has discussed the various reasons for why Gibrat's Law may be valid, as well as the factors that contribute to rejecting it. Economy-wide and firm-specific effects can aid in both rejecting and accepting the random walk Gibrat describes, depending on if the effects explain the variance or level of firm size. Later in this paper, we include the exchange rate (economy-wide) and the debt level (firm-specific) as variables that explain the size of campsites.

There are statistical and econometric challenges to testing Gibrat's Law (Novoa 2011). When using dynamic panel data analysis, the first choice is that of the dependent variable. There are essentially two alternatives: firm growth and firm size. By choosing growth, one takes the first difference of size, while using size as the explanatory variable (Oliveira and Fortunato 2008). In this case, Gibrat's Law holds if the parameter for size is insignificant. Alternatively, using size as the dependent and explanatory variable, the following model is applied:

$$y_{it} = \alpha + \beta y_{i,t-1} + \varepsilon_{it}, \qquad (1)$$

where $y_{it}$ is the logarithmic value of size for the actual company in a specific sector at time t. The lagged dependent variable is the only explanatory variable, $\alpha$ is a constant and $\varepsilon_{it}$ is random disturbance term. In this model, Gibrat's Law holds if it is shown that firms follow a random walk; that is, if $\beta = 1$. Deviations from this random walk give insight into the distributional trend of the sector's firms.

### 3.1. The Hypotheses Connected to Gibrat's LPE

If β > 1, the sector has an explosive trend, where larger firms grow proportionally faster than smaller firms. If this explosive trend persists over time, there will be few companies left, and the sector will convergence to oligopoly or monopoly. In young industries, the explosive trend could be relevant as an initial edge can exacerbate itself in succeeding years. This cannot last forever, however, which is one reason why older sectors tend to not reject the LPE. If β < 1, there is a mean reverting trend in the sector, where the mean growth is stronger among small companies as they are in a state of 'catch-up'. Thus, we can assume there exists a 'natural' or 'perfect' firm size at which firms will eventually return if they diverge from it, their long-run growth being equal. The steady state size of each firm need not be the same, but the firms will revert to some mean. In the extreme case, where β = 0, every deviation from this mean will be cancelled out in the next period, meaning that firms do not deviate for more than one period. In this case, current size is no predictor of future size.

Many papers (Novoa 2011; Oliveira and Fortunato 2008; Shehzad et al. 2009) have tested Gibrat's Law by following the procedure of Tschoegl (1983), which is a stronger version of Gibrat's Law that suggests three propositions (P1–P3). From Equation (1), we can write the growth for company *i* as:

$$y_{it} = \alpha + \beta y_{i,t-1} + \varepsilon_{it}, \text{ where } \varepsilon_{it} = \rho \varepsilon_{it-1} + u_{it}, \text{ and } u_{it} \sim N(0, \sigma^2) \qquad (2)$$

The sum of the error term's ($u_{it}$) deviation (σ) is by construction equal to zero. In addition, the variance of the error term is written as:

$$\sigma^2_{it} = \delta y_{it} + \eta_{it} \qquad (3)$$

which gives the three propositions (the null hypothesis) as:

P1: β = 1
P2: ρ = 0
P3: δ = 0

First (P1), the relative growth of each firm is independent of the firm's initial size, and firms follow a random walk. This is the firm size's autoregressive process. Second (P2), if a firm deviates from its growth path in one year, this deviation does not carry over to the next year; that is, extraordinary success/failure in one year does not translate into extraordinary success/failure the next year. The second proposition differs from the first in the sense that the first proposition concerns the firm's trend, whereas the second concerns deviations from this trend. This (P2) is equivalent to the firm size's moving average process. The third proposition (P3) states that the relative variance in firm size is independent of the firm's initial size. That is, small firms do not vary relatively more or less in their size than large firms. This is equivalent to the firm size's heteroscedastic property.

As stated, P1 holds if the firms collectively follow a random walk, meaning if the best prediction of future size is the current size and all deviations from this size follow a random process.

If P2 holds, all outside effects on the firm size for a given year will be completely reflected in the firm size for this particular year, and those effects will have no impact for the firm's growth in the coming years. A success one year will give an increase in the size of the firm, but this increase does not necessarily lead to a further increase in the next year. That is, the deviation in growth in one year will not carry over into the next year. This does not mean that the success/failure of one year disappears the next year, only that its effect is absorbed that year. If ρ = 0, there is no spill-over effect, and growth will normalize to the prior regular growth after an initial shock. If ρ < 0, firms with extraordinary success in one year will have considerably worse results the next year, with a growth below the average. A lucky period is followed by an unlucky following period, and vice versa: a failure one year results in good performance the next year, with growth stronger than the

others. If ρ > 0, extraordinary growth one year will persist into the following year. That is, growth over the average level for a given firm will persist into the following period. If a company has extraordinarily strong growth one year, it will also manage to maintain this above-average growth in the following period. If one year turns out badly for a company with a low growth, this will also yield negative consequences the following year. If firm growth is characterized by ρ > 0, we can view this a persistence in firm success/failure, or 'slowness' in firm growth. On the other hand, ρ < 0 can be viewed as success/failure being 'cancelled out'. Particularly for campsites, P2 can go both ways, depending on whether the visitors are (dis)pleased with more/fewer other visitors at the same campsite.

If P3 holds, the proportional variance of revenue for the companies is independent of their size—that is, firm size is not related to growth volatility. A negative δ value means there is a negative relationship between a firm's growth volatility and its size: smaller firms have relatively more volatility in their revenue stream. One interpretation is that smaller companies experience greater uncertainty than large enterprises, perhaps because smaller firms are more sensitive to consumer tastes and market conditions, whereas larger firms have a more stable revenue stream.

A study by Calvino et al. (2018) concluded that the value is negative, and remarkably stable across 21 selected countries. Goddard et al. (2004) investigated whether the previous year's growth has an impact on the actual growth, and found a positive relationship, but with no significant impact.

Many researchers have included independent and control variables to the estimators to see how this affects growth. By extending the model with other variables, one can test and explain how different factors contribute to growth and analyse why Gibrat's Law is rejected (Oliveira and Fortunato 2008). For instance, Donati (2016) showed how liquidity constraints limited the growth of small firms. Debt leverage as a control variable yields a mixed result (Jang and Park 2011; Phillips 1995). Some report a negative relationship (Billett et al. 2007), because higher debts increased the number of poor projects. On the other hand, a higher debt level can increase firm performance through successful ventures—that is, the level of debt can be seen as risk-taking.

### 3.2. Econometric Methods

In early empirical testing of Gibrat's Law using econometrics, the ordinary least squares (OLS) method of estimation was used. Due to the presence of the lagged dependent variable, this induces endogeneity issues through the feedback, or looping, mechanism, as shown by Chesher (1979). Consequently, as Chesher (1979) and Jang and Park (2011) have pointed out, this means that OLS will be inconsistent unless the number of variables representing firm size is equal to the number of time periods. When there are more than a few time periods, this becomes, at a minimum, inefficient, and infeasible at most. Even so, many researchers have used OLS to test Gibrat's Law (Daunfeldt and Halvarsson 2015). This is true for the previous studies that have analysed the validity of Gibrat's Law in camping sites (Italian and Dutch).

An alternative approach is to use the generalized method of moments (GMM) and, specifically, those methods that are specifically created for dynamic panel data scenarios. Arellano and Bond (1991) proposed such a method for panel data to ensure a consistent evaluation of the parameters. They exploited the moment conditions of the first differenced error terms, which allowed for the use of the lagged level of two periods prior as instruments for the first differenced equation. The estimator has been called the AB or FD GMM method. Some researchers have used it to test Gibrat's Law (Ivandić 2015), but when the autoregressive parameter (β) approaches unity, the instruments used become weaker. In the case that Gibrat's Law holds, β = 1, the instruments are entirely invalid, as they are not correlated with the first differenced equation. This leads to inconsistent and downwardly biased estimators of β, as has been shown in several studies using Monte Carlo simulations (Blundell and Bond 1998; Jang and Park 2011; Moral-Benito et al. 2019). As a result, using the Arellano–Bond estimator will tend to lead to a rejection of Gibrat's

Law too often. Due to the difficulties of the first difference GMM estimator (Arellano and Bond 1991), Arellano and Bover (1995) and Blundell and Bond (1998) developed an improved version of the dynamic panel data GMM estimator, which combines the lagged level instruments for the differenced equation with differenced instruments for the level equation. The method has been called the system GMM (SYS-GMM) estimator and has proven to be a very powerful dynamic panel data estimator, even when the autoregressive parameter β approaches unity. This is what the testing of Gibrat's Law requires, and it has been used in many articles (Donati 2016; Giotopoulos 2014; Jang and Park 2011; Oliveira and Fortunato 2008). In addition, due to the information contained in the level equation, one can estimate time-invariant variables, such as location and sector, which is not possible with the FD-GMM estimator. The System GMM estimator is the estimator of choice in a dynamic model with short panel data.

There are two main arguments against the SYS-GMM estimator, however. First is the issue of using instrumental variables, as they will always run the risk of becoming weak (as with the FD-GMM estimator). This risk is circumvented to a degree by the SYS-GMM estimator by using sets of equations with two sets of instruments, in addition to the fact that the instruments are the 'same' variable as those being instrumented. The instruments do require the sacrifice of a time period, which can be crucial in short panels. The second contention is more serious, as, contrary to the FD-GMM estimator, the SYS-GMM estimator requires the assumption of mean stationarity of the cross-sectional observations (firms). That is, by including the first difference instrument for the level equation, it assumes that all cross-section observations have reached a steady state (Allison et al. 2017; Moral-Benito et al. 2019). This translates into the assumption that each of the firms has reached their natural size and are in a steady state at the beginning of the sample period, which seems to be an unrealistic assumption for the camping sector (and many other sectors for that matter). Additionally, the moment conditions that both GMM estimators rely on require that there is no second order autocorrelation for validity of the instruments. If this does not hold, trice-lagged instruments need to be used, which would induce weaker instruments, and the sacrifice of another time period.

Consequently, Allison et al. (2017) and Williams et al. (2018) developed a maximum likelihood estimator based on simultaneous equations as an alternative to the GMM estimators; this is called the ML-SEM (maximum likelihood structural equation modelling) estimator. This is a computationally intensive method, which avoids both the instrumental variables issue of GMM, the limitations of FD-GMM, and the unrealistic assumptions of SYS-GMM. The ML-SEM estimator is slightly more precise and unbiased than the SYS-GMM estimator under a variety of conditions while relying on the same weak regulatory assumptions as the FD-GMM estimator, without using instrumental variables. For our study, the ML-SEM estimator's largest drawback is its novelty, as it is not fully optimized with our software, and we are not aware of any published article testing Gibrat's Law by applying ML-SEM.

## 4. Hypotheses

We use an extended version of Gibrat's Law, which contains three propositions that pertain to the sector's growth, inertia, and variance. This translates into testing the sectors: autoregressive, moving average, and heteroskedastic components, parameters which are obtained via dynamic panel data modelling. The three propositions translate into the three hypotheses, which encapsulate the stricter version of Gibrat's Law. The first hypothesis is:

**Hypotheses 1 (H1).** *Company growth is independent of their size ($\beta = 1$).*

This hypothesis is the original hypothesis of Gibrat's Law (P1), and it refers to the autoregressive process of the sector. If it holds, the firms follow a random walk. Due to the natural and government restrictions that the sector faces, we expect that the individual campsites have a 'natural' or steady-state size that they revert to in the long run. Therefore,

if H1 fails, we expect that the autoregressive component to be below unity for the sector size (equivalent to being below zero for sector growth).

The second hypothesis is one of the two additions of the stricter version:

**Hypothesis 2 (H2).** *Success or fiasco one year has no effect on growth in the subsequent year ($\rho = 0$).*

If this hypothesis does not hold, shocks to revenue are followed by additional shocks in subsequent years. That is, there is inertia in shocks to revenue.

The last hypothesis connected directly to Gibrat's Law is:

**Hypothesis 3 (H3).** *There is no link between growth volatility and firm size ($\delta = 0$).*

The third hypothesis is most often seen to fail, as small firms tend to have relatively higher volatility in size in comparison with larger firms.

Furthermore, we extend the model to investigate whether there is any link between the exchange rate, the level of debt, and the growth of campsites in Norway. This constitutes two models, the first one being represented in Equations (2) and (3). We may call this the restricted model, whereas the unrestricted model is as follows:

$$y_{it} = \alpha_{it} + \beta \, y_{i,t-1} + \gamma_1 \, x_{1t} + \gamma_2 \, x_{2,it} + \varepsilon_{it}, \text{ where } \varepsilon_{it} = \rho \varepsilon_{i,t-1} + u_t, \; u_{it} \sim N\,(0, \sigma^2) \quad (4)$$

where $x_{1,t}$ and $x_{2,it}$ represent the log of the exchange rate (NOK/Euro) and the log of debt, respectively. The exchange rate is defined as a predetermined variable, which means that it is not allowed to be affected by the other variables. This assumes that the campsite sector does not affect the exchange rate, which seems realistic for the Norwegian economic structure.

Recent research has shown that a depreciation in the Norwegian currency has increased the inflow of foreign visitors to Norwegian campsites (Idsø and Opstad 2021; Opstad et al. 2021a). We postulate the following hypothesis:

**Hypothesis 4 (H4).** *There is no correlation between currency rate and growth ($\gamma_1 = 0$).*

If H4 does not hold, this inflow of foreign visitors translates into higher/lower growth for the campsites, indicating that campsite revenue is sensitive to macroeconomic conditions. If this is the case, we would expect the variable to be positively significant.

Lastly, we include the debt level as a firm-specific variable with the hypothesis:

**Hypothesis 5 (H5).** *Firms' debts are not associated with growth ($\gamma_2 = 0$).*

Hypothesis five (H5) assumes that growth is independent of the level of firm debt. However, limited access to capital can prevent businesses from growing, which means that firms need capital to grow; again, we expect that if the assumption does not hold, it will be positively significant, indicating that firms that have more access to capital (through debt) have higher rates of growth.

First, we test these hypotheses for the whole sample, and then investigate further splitting the sample into three. By splitting instead of including size dummies and/or interaction terms, we can apply the complete analysis to each size segment without unnecessary complications.

## 5. Methodology
*The Sample*

The account information is taken from the Norwegian public register for firms (Brønnøysund Register Center). The population consists of 292 campsites from 2010 to 2019 (10 years). Campsites with zero or very low revenue during the period were excluded, and

so were campsites with two workers or less. All campsites in the sample have been in business for the entire time period.

The proportion of foreign visitors is about 25%. Due to the devaluation of the Norwegian currency (see Figure 1), more foreigners are visiting Norwegian campsites [45].

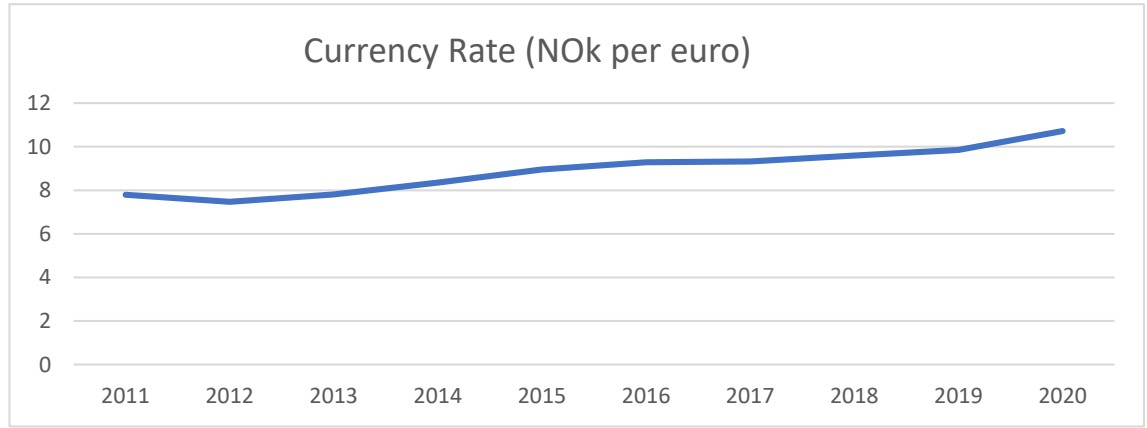

**Figure 1.** Exchange rate 2010–2020 (Source: Norwegian Central Bank).

Regarding the removal of campsites with few employees, these may indicate farmers or others who have a family-run enterprise in addition to other activities. Some of the removed 'campsites' were not in fact campsites, and there were missing data, uncertain figures, and considerable variation. We therefore removed all companies with two employees or fewer (see Table 1). Some other firms were excluded due to missing data or no revenue in at least one year. The final sample includes 176 campsites in Norway in a period from 2010 to 2019. The total observations thus total 1760. The companies are divided into three groups depending on the number of employees (Table 1)—small (3–9 employees), medium (10–25 employees), and large (more than 25 employees)—with, respectively, 81, 56, and 39 campsites in each of the groups. The larger campsites are mostly in the Norwegian southeast, these having more in common with the campsites of continental Europe. This contrasts with the northwestern campsites, which face harsher natural restrictions, as well as state regulations that serve to protect the unusual nature of these regions. Of the 81 small campsites, 54 are northwestern, whereas 23 of the 39 large campsites are southeastern. A total of 31 of the 56 medium-sized campsites are in the North West.

**Table 1.** Descriptive statistics with mean values, standard deviations in parenthesis, and min and max values in brackets.

|  | All | Removed [1] | Small Firms | Medium Firms | Large Firms |
| --- | --- | --- | --- | --- | --- |
|  | (*n* = 277) | (*n* = 97) | (*n* = 81) | (*n* = 56) | (*n* = 39) |
| Revenue (1000 NOK) | 4643 (7225) [0,84,336] | 1720 (3032) [0,20,548] | 2943 (1860) [466,9673] | 6416 (5968) [932,30,375] | 13,070 (13,819) [2598,84,336] |
| Employees No. | 10.4 (13.8) [0,82] | 0.5 (0.8) [0,2] | 5.7 (2.0) [3,9] | 24.9 (4.4) [10,25] | 38.7 (13.9) [26,82] |
| Debts (1000 NOK) | 4623 (6316) [15,70,160] | 3432 (4953) [15,32,396] | 3412 (3669) [72,22,906] | 5129 (5474) [85,30,004] | 9495 (11,089) [827,70,160] |

Notes: [1] Firms with under 3 employees. Mean, (st.dev), [Min,Max].

The applied statistical methods are the Arellano–Bond (FD-GMM), Blundell-Bover (SYS-GMM), and Moral-Benito et al. (ML-SEM) estimators. The dependent variable, size, is measured by the level of firm revenue in the period. The extended model is estimated

with the system GMM estimator, and so are the tests. Robust standard errors are used, and whereas the robust standard error properties for ML-SEM have not been investigated, there is no immediate reason to think that they are invalid, and they do not differ much from the robust standard error properties of the two other estimators.

## 6. Findings

Tables 2 and 3 present the results. Notice in Table 2, the three different estimator methods give almost the same result. With the sample used in this analysis, the choice of estimator method has little impact on the conclusion in the testing of Gibrat's LPE. Consequently, the focus will be on the estimates from the SYS-GMM in the subsequent discussion, as this is the most widely used estimator for these purposes.

**Table 2.** Dynamic panel data estimators testing Gibrat's Law for Norwegian campsites over a 10-year period (2010–2019) (Robust Standard error in parentheses).

|  | All | Small Firms | Medium Firms | Large Firms |
|---|---|---|---|---|
|  | (*n* = 176) | (*n* = 81) | (*n* = 56) | (*n* = 39) |
| **AB-GMM** |  |  |  |  |
| AR(1) | 0.8608 * | 0.6758 | 0.8480 ** | 0.8951 |
| (Auto Regression) | (0.071) | (0.237) | (0.065) | (0.079) |
| MA(1) | 0.067 | 0.0558 | 0.0174 | 0.1599 ** |
| (Moving Average) | (0.049) | (0.128) | (0.111) | (0.051) |
| **SYS-GMM** |  |  |  |  |
| AR(1) | 0.8533 ** | 0.6438 | 0.7721 *** | 0.855 |
| (Auto Regression) | (0.073) | (0.246) | (0.069) | (0.098) |
| MA(1) | 0.0736 | 0.0239 | 0.0092 | 0.1538 *** |
| (Moving Average) | (0.05) | (0.145) | (0.579) | (0.046) |
| Heteroskedasticity(t) | −9.20 ** | −10.41 ** | −14.35 *** | 0.31 |
| Autocorrelation(p) | 0.321 | 0.273 | 0.539 | 0.096 * |
| Cross-sectional dependence | 34.50% | 36.70% | 31.20% | 35.60% |
| **ML-SEM** |  |  |  |  |
| AR(1) | 0.8653 * | 0.6917 ** | 0.8532 ** | 0.915 |
|  | (0.077) | (0.132) | (0.065) | (0.08) |
| AR(2) | 0.0392 | −0.0195 | 0.212 |  |
|  | (0.052) | (0.18) | (0.058) |  |

Notes: AR(1) tests H1 ($\beta = 1$), MA(1): H2 ($\rho = 0$) and Heteroscedasticity(t): H3 ($\delta = 0$). Not possible to estimate AR(2) for large firms *** $p < 0.01$, ** $p < 0.05$, * $p < 0.1$.

**Table 3.** SYS-GMM with including control variables. Dependent variable: Growth.

|  | All | | Small Firms | | Medium Firms | | Large Firms | |
|---|---|---|---|---|---|---|---|---|
| **SYS-GMM** | Mod 1 | Mod 2 | Mod 1 | Mod 2 | Mod 1 | Mod 2 | Mod 1 | Mod 2 |
| AR(1) | 0.853 ** | 0.757 *** | 0.644 | 0.472 *** | 0.772 *** | 0.673 *** | 0.855 | 0.736 * |
| (Auto) | (0.073) | (0.052) | (0.246) | (0.182) | (0.069) | (0.078) | (0.098) | (0.126) |
| MA(1) | 0.0736 | 0.052 | 0.024 | −0.043 | 0.009 | −0.003 | 0.154 *** | 0.153 *** |
|  | (0.05) | (0.047) | (0.145) | (0.135) | (0.115) | (0.095) | (0.046) | (0.048) |
| Exchange Rate | | 0.382 *** | | 0.798 *** | | 0.462 *** | | 0.39 |
| (NOK per euro) | | (0.116) | | (0.256) | | (0.222) | | (0.324) |
| Debt | | 0.081 *** | | 0.109 | | 0.117 * | | 0.120 ** |
|  | | (0.035) | | (0.070) | | (0.062) | | (0.056) |

Notes: Mod 1 is without control variables (see Table 1). Mod 2 includes control variables. *** $p < 0.01$, ** $p < 0.05$, * $p < 0.1$.

Hypothesis H1 is rejected in favour of $\beta < 1$ for the whole sample with a significance level of 5%, and for medium campsites with a significance level of 1%. For large firms, we cannot reject the hypothesis $\beta = 1$; there is no evidence that they do not follow a random

walk. The coefficient is close, but below unity for this subsample. We cannot reject a random walk for the smallest firms either, but this is not due to the coefficient being close to unity. Instead, the standard error is too large. This indicates that the smallest campsites have large variations in their growth paths, where most revert quickly to their mean, whereas others may follow random walks, and still others may even have explosive growth paths. The same dynamic of small, medium, and large campsites, where there seems to be a threshold size where the campsites' path changes, is also reflected in the moving average parameter, the heteroskedasticity tests, and the autocorrelation tests.

Firstly, from the size-related heteroskedasticity test, we can see that, for the whole sample—and the small and medium firms—the variation in their size (revenue) decreases the larger the firms are. This indicates that there may be a threshold size for campsites, where additional size does not translate into more stable revenue.

We see the same threshold dynamic in the moving average component, where the only significant MA component is found in the large firms. The revenue-deviations of large campsites spill over from one year to another, one year's success being a significant predictor of success in the following year. The success or failure of large campsites in one year persists into the next year, whereas the success/failure of small and medium campsites is absorbed into their revenue in the year the success/failure happens. That is, shocks that occur to large campsites have a certain inertia as it pertains to their size (revenue). This is reflected in the test for second year autocorrelation, which is only significant for large campsites. Consequently, the moving average parameter, the heteroskedasticity tests and the autocorrelation test all point to there being a threshold size for campsites at which their growth paths change.

The absolute average value dependence across the sample is 34.5%, and it is quite stable regardless of the size of the campsites. One reason might be competition between campsites, whereas another is that they are all affected by the same market forces and consumer tastes. Model 2 includes two other independent variables (exchange rate and debt; see Table 2). Both have the expected positive signs, and both are significant at the one percent level. A depreciation in the exchange rate has been shown to lead to more foreign visitors, but it might also lead to more Norwegians choosing to stay in the country for their vacation.

## 7. Discussion

### 7.1. Hypotheses H1 to H3

The median firm sub-sample reflects the whole sample well. For all campsites, the value of $\beta$ equal to 1.0 is rejected at the 5% significant level (SYS-GMM estimates) in favour of $\beta < 1.0$. Consequently, there is a tendency for Norwegian campsites to revert to some mean, steady state, or natural size. This is in contradiction to previous studies of campsites, but not in contradiction to other studies using the SYS-GMM estimator with moderately long time dimension. The results suggest that smaller campsites are growing significantly faster than larger ones. For the smallest campsites, there is considerable variation in the estimates, and this gives an uncertain result. The picture is different for the medium-sized campsites, in which the smallest firms grow significantly more than the mean company in this segment. This result is not found for large campsites, indicating that after the campsites have reached a certain level (more than 25 employees), the growth for the individual enterprise is independent of its size. We can thus conclude that the growth of a company within the camping sector is also independent of its size after a certain size is reached. Because the campsites belong to a labour-intensive sector in which there are limited localization opportunities (e.g., near the sea or lake), the possibility of economies of scale may be limited, explaining the results for the mean reversion for the Norwegian camping sector as a whole. Consequently, the crux of Gibrat's Law, the random walk of firm size for the sector, is rejected in favour of mean reversion.

Notice also that H2 ($\rho = 0$) is only rejected for large companies in favour of $\rho > 0$ with a significance level of 1%. This is an interesting result. If a large campsite does something

extraordinary in a year that contributes to higher growth, this will keep going into the following year, which will show above average growth as well. That is, it has a spill-over effect for the next period, and therefore the company achieved in higher-than-average growth also for the subsequent period. The company will also have an advantage in subsequent periods based on the prior year's success. Similarly, a fiasco one year resulting in lower growth will persist from one period to the next. For the camping sector generally, however, there is no significant spill-over effect from year to year.

The volatility of companies in camping services is strongly dependent on size. With a negative significant value for the heteroskedasticity test, this means small campsites have a greater variation in growth than the average company in the sector. This confirms the results of other researchers with data from other sectors (Calvino et al. 2018; Coad 2008). There is noticeably more stability within large firms than among small ones. Smaller firms show greater fluctuation in their growth than the others. This is strongly the case for small- and medium-sized enterprises. The exception is for campsites with more than 25 employees; in this category, volatility does not depend on firm size. Volatility is highly dependent upon size, meaning that H3 is also rejected in line with existing literature.

Analysis from other sectors suggests this difference between large and small may be due to better use of technology, more differentiated activity that reduces risk and longer company history (Begenau et al. 2018). It is reasonable to assume that large campsites are more likely to have several units with different geographical locations and with a focus on multiple segments (year-round operation, winter and summer holidays, cabins, campers) and various types of activities (family activities, hiking, fishing, etc.) aimed at both domestic and foreign visitors. In this way, the risk is spread out and the firm is less vulnerable if there is a decline in any individual field (bad weather, a reduction in foreign visitors, etc.). Smaller companies cannot spread out their activity and risk in the same way and therefore become more vulnerable, which causes greater fluctuations. The larger corporations seem to be working on a longer time scale, a dynamic that is often observed both in economics and in nature.

### 7.2. Other Explanatory Variables (H4 and H5)

A fall in the Norwegian currency rate contributed to increased visits by foreigners. It is therefore no surprise that this caused increased growth in the camping sector. However, this does not apply to campsites with more than 25 employees. One possible explanation is that the large companies have such a diversified portfolio of activities that they are less vulnerable to the inflow of foreign visitors. They may also have a supply that has a more inelastic demand regarding fluctuations in the exchange rate. Large enterprises may also adjust prices to compensate for changes in demand due to such exchange rate fluctuations. Alternatively, foreign visitors could be visiting small- and medium-sized campsites disproportionately more than they visit large campsites. Either way, H4 is rejected in favour of a deprecation leading to higher revenue, except for the large campsites, whereas small campsites are more sensitive to the exchange rate than medium-sized campsites. A weaker Norwegian currency increases growth in the camping sector.

As for H5: A 1% increase in the level of debt in one year is estimated to translate into a 0.08% increase in the revenue in the same year for the entire sample, when controlling for the other variables. Although this effect may seem small, it must be remembered that we have controlled for the previous year's revenue, and debt levels might be expected to work on a larger timescale and/or with a certain time-lag. Even then, the results are telling, especially when looking at the sub-samples. In essence, the larger the campsite is, the more significant the level of debt is in explaining its level of growth. This indicates that larger campsites are more successful in turning investments into revenue than smaller campsites, which can either be due to smaller campsites being less experienced in carrying out projects, or that their investments fall through more often due to their sensitivity to market forces or consumer tastes.

## 8. Conclusions and Contribution

The crux of Gibrat's Law is that the best prediction one can make about future firm size is current firm size. This is the starting point of Gibrat's Law, whereas the stricter version adds to more hypotheses. Firstly, all deviations from this initial size come from a white noise error term, and secondly, the variance of this error term is independent of size. If all three hypotheses hold, we can say that the firms follow 'pure' random walks.

This is not the case for Norwegian campsites, in contrast to Italian and Dutch campsites, but in line with most other sectors and industries. We find that the size of Norwegian campsites is mean reverting, and its volatility decreases with size. Hypotheses 1 and 3 are thus rejected, the firms converge towards a 'natural' steady state, and they become more stable as they grow. Gibrat's Law does not hold for Norwegian campsites.

Furthermore, we find evidence of a threshold size for the Norwegian campsites, at which point their growth processes switches. At the point of about 25 employees, the distinction between the medium and large sub-samples, the processes change. When the threshold size is reached, there is no longer any gain of increased size in the stability of growth, and the current success/fiasco becomes a predictor of future success/fiasco. In addition, we can no longer reject a random walk after this threshold size. Consequently, hypothesis 2 is rejected for the large Norwegian campsites, whereas hypotheses 1 and 3 are not. This is the opposite result of the general result we obtained for all Norwegian campsites, and more in accordance with previous studies of campsites in Italy and the Netherlands.

As for hypothesis 4, we can see that a depreciation of the Norwegian Krone translates into higher revenue for the sector, as more foreign tourists choose to visit the country, whereas fewer domestic tourists choose to leave the country. The differing degree to which the exchange rate affects the three sub-samples is grounds for further research.

Hypothesis 5 shows that higher leverage leads to higher revenue streams, but not for small campsites. This can be due to an unwillingness or inability to invest or gain the means to do so. Whether the level of debt is positively related to profitability is another issue, investigated by Opstad et al. (2021b).

We used three estimators to obtain the autoregressive and moving average components of Norwegian campsites, the FD-GMM, SYS-GMM, and ML-SEM estimators. Although they have differing strengths and weaknesses, the results were similar. The steady state assumption of the SYS-GMM seems to not cause too many problems, comparing it to the other two. The Monte Carlo evidence against the FD-GMM estimator when the autoregressive component approaches unity would seem to make it inappropriate for testing Gibrat's Law, although our study does not show it conclusively. The ML-SEM estimator, combining the weak assumptions of the FD-GMM estimator with better precision than the SYS-GMM estimator, seems to be the best choice for testing the dynamic properties of firms, according to Monte Carlo evidence. We are not aware of any studies using it to investigate Gibrat's Law in the literature yet, this being an introduction of the estimator to the literature.

To conclude, another novel contribution of our paper to the general Gibrat's Law literature, is the evidence of a threshold size, for the tourism industry at least. We show evidence of this threshold size (non) rejection of hypotheses 2 and 3, and to a lesser degree hypothesis 1. At the point of 25 employees, in the case of Norwegian campsites, size no longer translates into more stability for the firm, but rather into a spill-over dynamic where current success/fiasco is carried over into the next year.

## 9. Limitations and Further Research

The data analysed were limited to 10 years and from only one country, and were also based on public statistics (from the Brønnøysund Register Center), thus some information from individual campsites that would have been of interest (e.g., prices) is lacking. There is limited research that applies such analysis to campsites, which limits the ability to compare

the present results with other findings, but we hope this article can be an important contribution in helping to explain the growth in campsites.

Furthermore, there is no data available to differentiate the different types of campsites in the sample. Analysis on how different types of campsites grow, and in which specific market they operate (sightseeing, exploring, pitstop) and which customers they specialize in (domestic, foreign, one time-or yearly visitors) are fields in which further research can be carried out. Additionally, access to data from campsites in other countries can help investigate whether our differing results are due to differences in country-specific factors, or due to methodological differences. A comparison of campsites across countries, or a comparison of different sectors in the Norwegian tourism industry are obvious paths to take for further research. Inclusion of additional macroeconomic variables, such as the exchange rate, can give more answers that are relevant for the tourism industry of all countries, as tourists can only go one place at a time.

Our analysis is limited in that the data at hand only contain firms that have been active for the entire sample period. Including entry/exit into the analysis could shed light on the high variance of the growth paths of the small firms. For instance: what decides which newly created campsites converge towards the steady state, and which fail? Accordingly, why do some campsites follow (potentially) explosive paths, whereas other campsites almost follow white noise paths? Another choice could be to include age as an explanatory variable. These points will help us to learn more about the life cycle of firms, where some become successful businesses, but whereas most die out.

**Author Contributions:** These authors contribute equal to this work. All authors have read and agreed to the published version of the manuscript.

**Funding:** This research received no external funding.

**Institutional Review Board Statement:** Not applicable.

**Informed Consent Statement:** Not applicable.

**Data Availability Statement:** Public statistics.

**Conflicts of Interest:** The authors declare no conflict of interest.

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
