# Peer review of "Evidence of a Threshold Size for Norwegian Campsites and Its Dynamic Growth Process Implications—Does Gibrat’s Law Hold?"

_economies, doi:10.3390/economies9040175_

Round 1
Reviewer 1 Report
This paper tests Tschoegl’s (1983) stronger version of the Gibrat’s Law for companies of different sizes within the Norwegian campsite industry. This version involves the fulfilment of three conditions, namely, firm growth is independent of firm size, the deviation in growth in one period does not carry over into the next period and firm size is not related to growth volatility. Two dynamic panel data models are estimated to this aim. Unlike previous results in the literature, it is found that Gibrat’s Law does not hold for the campsite industry.
This is a well-written paper that I enjoyed reading. However, I wonder why it is relevant to study whether Gibrat’s law holds for the camping industry. The authors do not provide a clear motivation for their work. Previous studies on this issue do. Indeed, as Gibrat’s law does not hold for manufacturing, Audretsch, Klomp, Santarelli, and Thurik (2005) posed the question on whether services are different. Piergiovanni, Santarelli, Klomp and Thurik (2003) tested if new-born firms and incumbents have a different behaviour regarding Gibrat’s law. Both studies found that Gibrat’s law holds for the camping sector. Why do the authors obtain different results? Do they consider a different version of the Gibrat’s law? Do they apply a more accurate methodology? Did previous studies pass over a relevant aspect? The authors should convincingly answer these questions to motivate their paper.
Additional comments are listed below.
- The abstract should be rewritten to include the motivation, hypotheses to be tested, methodology (dynamic panel data models) and findings. Regarding findings, the authors must describe them clearly. Note that the title is a question, so the authors must provide an answer.
- In the abstract, the authors states Gibrat’s law is partially rejected. What does it mean? The law must be either accepted or rejected. In this respect, the authors must refer to the hypotheses instead of Gibrat’s law.
- In the introduction, the motivation of the paper (now missing) must be stated clearly and convincingly.
- Section 3 is very well drafted.
- On line 233, the model (in equation 4) includes the log of the exchange rate and the log of debt. However, on lines 323-324 the authors point out that “Model 2 includes three other independent variables (exchange rate, debt and wage share; see Table 2).” The variable wage share is absent in the analysis.
- The appendix repeats some information already presented in sub-section 3.2.
Author Response
Thank you for your comments, they have helped us improve our paper. We have improved the presentation of the results, making them clearer. The abstract and the introduction have been changed according to your remarks. In response to your additional remarks:
- A paragraph has been added to explain the relevancy of our study (line 34-38).
- The phrase “partially rejected” has been rewritten in the abstract.
- Further motivation for the paper has been included in the introduction.
- Mentions of wage have been removed.
- The appendix has been incorporated into section 3.
Reviewer 2 Report
- The article is too difficult to read and to understand.
- I suggest revising the title to reflect the content of the paper in a more accurate way. For example a general overview and the an specific sentence.
- The abstract is well articulated: introduction, research gap, methodology, paper's structure, results and contribution to the academia.
- Keywords: please include Norway. Remove Sys-GMM estimator; dynamic panel data
- Abstract line 4-5: please specify the study area.
- Introduction: please clearly identify the literature gap your research aims to fill in.
- Line 38-39 please sustain in stadistics the sentence "the tourism industry.... contributes to value creation and enployment. Please use data from UNWTO or ILO.
- Line 40-41 please sustain also in stadistics the sentence "campsites are important contributor...."
- Hyphoteses: please explain and sustain in literature the hyphoteses. In the introduction, the author/s mention the size has not effect, but then the size is includes as hyphotesy.
- Methodology: please add information about the location of the campsites and the category (from my point of view these two parameters could be relevant to explain the results).
- The methodology approach is too vague. It would be interesting to include other methodologies, such us interviews or focus group to achieve a triangulation.
- Findings: is H5 rejected or not? Please mention it between the lines 334-344.
- Discussion: the author/s should extend the discussion with the authors mention in the literature review.
- The discussion is expected to develop the relationship between results and literature from a more critical perspective.
- When discussion the results, please highlight the theoretical contribution.
- The discussion of the hyphoteses has to be separated. I mean, the author/s can not grouped (i.e H1 to H3)
- Limitation should be included in the section "Conclusions and further research".
- Do not consider the category of the campsite in the research it is a limitation. Please include it.
- The conclusions should be extended. The practical application of the research (the management implications) presented in the conclusions are vague and brief. Perhaps more can be said about the potentially impact and the consequent tourism experience management?
- The reference list should be arranged alphabetically.
- The appendix should be included in the section "methodology".
- From my point of view, the author/s should reestructure the article. I can not see the practical contribution of the article
Author Response
Thank you for your criticism, they have helped us improve out paper. Following your remarks, we have restructured large parts of the article, and made it easier to read. A point-by-point response follows:
- The title has been changed.
- Norway has been included in the Keywords.
- The introduction has been revised, now including the literature gap: with focus on campsites.
- The requested lines (40-41) have been removed.
- The hypotheses have been split, specifically in part 3.1 (H1-H3, Gibrat’s LPE) and part 4 (H4-H5).
- Location and size of campsites has been added (line 360-361).
- We agree that including other methodologies, such as interviews and focus groups, would be interesting, but this was not available for this article. This can be followed up in later projects.
- Detailed discussion of H5 has been added in section 7.2. Generally, the discussion has been extended.
- The category of campsites has been included in “Limitations”, as we have no data for this variable.
- The conclusion has been thoroughly extended.
- The reference list has been made alphabetical.
- The appendix has been incorporated into section 3.
Reviewer 3 Report
Overall the paper is not bad at all. The hypoteses connected to Gilbrat's LPE are well defined. The econometric problems are clearly explained. The chosen statistical methods are correctly applied, although there are some questions (see below). The results are correctly commented. Overall the paper is well written (apart very small inaccuracies and few typos).
However, the aims behind the work are not clear. The reasons that make the problem relevant are not explained. Why is important to test the Gilbrat's law in the Norwegian campsite sector? Why is it important to explain the structure and concentration in the (Norwegian) campsite case? Are there any context-specific problems that suggest the Gilbrat's law should be tested? (e.g. labour legislation constraints, environmental problems, shortage of supply, etc.). Which sector structure is desirable considering these problems? Ect. In other words, what is the foundamental purpose of the work? Only to "fill a hole" (no test of Gilbrat's law in Norwegian campsites)? Or compare different statistical methods to test the "law"?
Coming to the curiosities related to the statistical models, the first one concerns the "separation" of the sample in three subsamples (Small, Medium and Large firms). Whas it not possible to include these characteristics in the model as explanatory variables? I also wonder about identification issue in the weak instrument problem of the system GMM estimator in dynamic panel data models. There are methods to assess these questions (see https//:www.jstor.org/stable/23116949?seq=1#metadata_info_tab_content).
There is also some confusion at page 7. If the authors consider mainly SYS-GMM, for ALL, Hypothesis 1 is rejected with a significant level of 5%.
Finally, at the end of page 9 there is the term "echnology".
Author Response
Thank you for your positive review of our article. Your comments have helped us greatly. In the introduction, we have included reasons for why our research question is relevant for the Norwegian camping sector.
Your questions about our methodology have been especially enlightening. The separation into three sub-samples have been explored further, and we have included reasons as for why it is necessary. In short: by dividing the sample instead of including the separation as dummy variables, we are able to distinguish the effects more clearly. In particular, the interactive terms needed would quickly to obtain the same parameters would quickly become messy.
The weak instrument issue you bring up is very good, and we will surely include it in further research. Unfortunately, the code I used was not compatible with the tests you requested, and I did not have enough time to learn/apply the code needed. Either way, I see your point completely.
The confusion on page 7 has been sorted out, and the typo on page 9 is fixed.
Round 2
Reviewer 1 Report
The work has improved substantially. I have really enjoyed reading this new version. However, I still think that the authors need to motivate their analysis more clearly. This is not a difficult task, since the main aspects to be highlighted are already present in the new version of the paper, but not in the Introduction.
My comments are listed below.
- In the Abstract, the authors must mentioned the methodology used in the analysis.
- I see two aspects that should be strongly highlighted in the Introduction to motivate the paper. First, unlike previous literature, the present paper uses an accurate methodology to test Gibrat’s law (as explained in Section 3.2). This should be clearly and summarily explained. Second, the study of industry dynamics is not only of interest from an economic policy point of view, but also for private investors (last sentence in Section 9). The authors should explain why this is so.
- There are a few typos in the paper that need to be fixed. For instance, on line 39 it should say “Gibrat’s Law” instead “Gibrat Laws”.
- On lines 143-145, the sentence “In this model, Gibrat’s Law holds if it is shown that firms follow a random walk; that is, if β =1, Gibrat’s Law is confirmed, and the firms follow a random walk, growth being independent of size” sounds fairly reiterative. It must be rewritten.
- In equation (2), there is a notational error. Variable μit does not exist.
Author Response
Thank you for your continuous constructive critisicm. We are happy you enjoyed reading the paper.
- The methology is mentioned in the abstract.
- The introduction has been supplemented with your positive observation of our paper. Your query about private investors is valid, but we have decided to remove this sentence from section 9.
- The minor errors you point are accurate, and we have fixed them. Thank you.
Reviewer 2 Report
First, I would like to acknowledge the authors' efforts in significantly revising the manuscript. In this revision, the research gap is better articulated and the practical contribution is highlighted. Therefore, I still see some room for further development. My comments are as follow:
- Introduction: the practical contribution of the paper has to be highlighted.
- Sorry but I do not understand the section "additional hyphoteses".
- The five hyphoteses has to be described in the same section.
- Please do not group the hyphoteses and present each of them separately.
- The section "limitation and further research" should appear after "conclusion and contribution".
- Overall, the revised manuscript is clearly written.
- From my point of view, the article lacks conceptual contributions.
Author Response
Thank you for your feedback. Your comments have helped us improve our article.
- The practical contribution of our paper has been included in the introduction.
- The hypotheses have been structured according to your suggestion. Thank you.
- Limitation and further research have been placed after conclusion and contribution.
In the introduction, we have attempted to emphasize our contribution further (line 31-32).
Round 3
Reviewer 2 Report
In my review of the original submission, I felt the manuscript was not particularly original. While the revision is an improvement, I still feel that the paper lacks conceptual contributions.